# Cardiometabolic and Nutritional Morbidities of a Large, Adult, PKU Cohort from Andalusia

**DOI:** 10.3390/nu14061311

**Published:** 2022-03-21

**Authors:** Elena Dios-Fuentes, Montserrat Gonzalo Marin, Pablo Remón-Ruiz, Rosa Benitez Avila, Maria A Bueno Delgado, Javier Blasco Alonso, Viyei Kishore Doulatram Gamgaram, Gabriel Olveira, Alfonso Soto-Moreno, Eva Venegas-Moreno

**Affiliations:** 1Unidad de Gestión Clínica de Endocrinología y Nutrición, Instituto de Biomedicina de Sevilla (IBiS), Hospital Universitario Virgen del Rocio/CSIC/Universidad de Sevilla, 41013 Sevilla, Spain; mariae.dios.sspa@juntadeandalucia.es (E.D.-F.); r.beniteza@hotmail.com (R.B.A.); alfonsom.soto.sspa@juntadeandalucia.es (A.S.-M.); evam.venegas.sspa@juntadeandalucia.es (E.V.-M.); 2Instituto de Investigacion Biomédica Malaga, Servicio Endocrinología y Nutrición, Hospital Regional Universitario Malaga, Universidad de Malaga, 29010 Málaga, Spain; montsegonzalo@yahoo.es (M.G.M.); viyeyk.doulatram.sspa@juntadeandalucia.es (V.K.D.G.); gabrielm.olveira.sspa@juntadeandalucia.es (G.O.); 3Unidad de Metabolopatías, Unidad de Gestión Clínica de Pediatría, Hospital Universitario Virgen del Rocio, 41013 Sevilla, Spain; mariaa.bueno.sspa@juntadeandalucia.es; 4Unidad Gastroenterología Pediatrica, Servicio Pediatria, Hospital Regional Universitario Malaga, Instituto de Investigacion Biomédica Malaga, Universidad de Malaga, 29010 Málaga, Spain; javierblascoalonso@yahoo.es; 5Centro de Investigación Biomedica en Red-Diabetes y Enfermedades Metabólicas Asociadas (CIBERDEM), 29010 Málaga, Spain

**Keywords:** phenylketonuria, sapropterin, phenylalanine

## Abstract

The establishment of national neonatal screening systems has resulted in improved quality of life and life expectancy in patients with phenylketonuria (PKU). This has led to the development of multidisciplinary treatment units for adult patients with PKU. We present a retrospective descriptive study of a cohort of 90 adult patients (>16 years) with PKU under active follow-up in two reference centers in Andalusia. We analyzed disease severity, treatment type, demographic variables, cardiovascular risk factors, vitamin and hormone profiles, and bone metabolism. The median (interquartile range)age was 29 (23–38) years, 47 (52.2%) were women and 43 (47.8%) were men. Eighty (88.9%) had classical PKU, five (5.6%) moderate PKU, and five (5.6%) mild PKU. Diagnosis was by neonatal screening in 62 (68.9%) of the patients. The rest had late diagnosis. Treatment with sapropterin was given to 18 (20%) patients and diet and nutrition therapy to 72 (80%). There was adequate metabolic control according to Phe levels in 43 (47.78%) patients. Body mass index was 26.61 (22.7–31.1) kg/m^2^. Twenty-six (29.2%) patients had obesity, 7 (7.9%) hypertension, 2 (2.2%) type 2 diabetes, 26 (28.89%) dyslipidemia, 14 (15.6%) elevated total cholesterol, 9 (15.8%) decreased high-density lipoprotein cholesterol and 16 (17.8%) hypertriglyceridemia. Seven (10.3%) patients had osteoporosis and 28 (41.17%) osteopenia. Twenty-six (30.6%) had vitamin D (25OH) deficiency and four (4.5%) vitamin B12 deficiency. Although we observed no differences with most vascular risk factors, we found a high prevalence of obesity in relation to the age of the cohort. A continued evaluation of comorbidities in these patients is therefore needed, despite adequate metabolic control.

## 1. Introduction

Phenylketonuria (PKU) is an autosomal recessive disease secondary to an alteration in phenylalanine metabolism, and is the most common inborn error of amino acid metabolism. The cause of the disease lies in a functional deficiency in the phenylalanine (Phe) to tyrosine (Tyr) step due to impairment of the enzyme phenylalanine hydroxylase (PAH). This enzymatic step occurs mainly in the liver. A wide range of mutations have been described in the PAH gene (12q22-q24.2), which encodes this enzyme.

In the absence of neonatal diagnosis and adequate treatment, symptoms of the disease appear in the first months of life and stem from the pathological accumulation of Phe, a metabolite toxic to the central nervous system, and from secondary Tyr deficiency. This leads to irreversible intellectual deficit, motor deficits, eczematous rash, autism spectrum disorders, epilepsy, developmental problems, behavioral disturbances, and psychiatric symptoms. Although the pathophysiology of brain dysfunction is still unclear, Phe levels are related to cognitive development [1,2].

Neonatal screening allows for the early diagnosis of PKU, improving clinical outcomes, life expectancy, and quality of life in PKU patients [3,4]. Screening is performed using tandem mass spectrometry to measure blood Phe and Tyr levels 48–72 h after birth. The prevalence of the disease in Spain is 1:6500 births, while the estimated prevalence in Europe is 1:10,000 births [2]. In Andalusia, PKU neonatal screening began in 1978.

There is no consensus for the classification of PKU, but according to the European guidelines on phenylketonuria, it can be categorized based on three criteria: pre-treatment plasma Phe concentrations; the degree of daily Phe tolerance in the diet; and the potential response to tetrahydrobiopterin (BH4). According to pre-treatment Phe levels, PKU is classified as severe or classical, moderate, mild, or benign hyperphenylalaninemia [2,3].

The main goal of treatment is to reduce blood Phe levels to achieve adequate neurodevelopment in children and normal neurocognitive and psychosocial functioning in adults, while ensuring optimal physical growth and nutritional status.

The mainstay of treatment is diet and nutrition therapy, which is based on a diet restricted in natural proteins, supplemented with hydrolyzed formulas of Phe-free amino acids and enriched in Tyr, vitamins and trace elements, with the aim of reaching 100% of the protein and caloric requirements needed, adapted to each stage. Follow-up and treatment should be continued for life.

Adult patients with PKU treated from infancy may present several types of complications in follow-up: those related to poor control of Phe levels (mainly neurological or neuropsychological, such as cognitive functioning, psychosocial function, mental health, Parkinsonism, neurodegenerative diseases, etc.); those that occur as a result of dietary restriction of proteins, vitamins or minerals (nutritional status, bone mineral health, etc.) [5]; and diseases of adulthood (fertility, obesity, diabetes mellitus, hypertension, cardiovascular events, dyslipidemia, etc.). In cases where adherence to nutritional supplementation has been inadequate, an increased risk of micronutrient deficiencies such as iron, zinc, selenium and vitamin B12 deficiency have been observed [2].

Alternative treatment options are available for PKU. In some instances, treatment with sapropterin dihydrochloride, which is a synthetic analog of natural BH4 and acts as a cofactor that increases residual PAH activity, can be effective in reducing Phe levels, improving protein tolerance, and allowing less restrictive diets to be prescribed in these patients.

This study aimed to describe a cohort of adult patients with PKU treated in Andalusia, examining clinical variables, metabolic control and the presence of nutritional, cardiovascular and bone mineral health complications.

## 2. Materials and Methods

This was a retrospective descriptive study of a cohort of adult patients (>16 years) with PKU under active follow-up in Andalusia (total population 8,476,718 million inhabitants). Included were all adult patients under follow-up between 2019 and 2020 in the Inborn Errors of Metabolism Units of the Endocrinology and Nutrition Clinical Management Units of the Virgen del Rocío University Hospital in Seville and the Regional Hospital of Malaga, which serve the total Andalusian population with errors of metabolism. A total of 90 patients with PKU were analyzed, 64 from Virgen del Rocío University Hospital and 26 from the Regional Hospital of Malaga. Patients with benign PAH and dihydropteridine reductase deficiency were excluded from the study.

According to pre-treatment Phe levels, patients were classified into classical PKU (>1200 µmol/L), moderate PKU (600–1200 µmol/L) or mild PKU (360–600 µmol/L). Good metabolic control was defined as Phe levels below <600 µmol/L according to the criteria of the European Society for Phenylketonuria.

As cardiovascular risk factors, lipid profile, the presence of diabetes mellitus, hypertension, smoking, and sedentary lifestyle were examined, in addition to assessing body composition, body mass index (BMI), and homocysteine levels. Hypercholesterolemia was defined as total cholesterol >200 mg/dL, high-density lipoprotein (HDL) cholesterol <35 mg/dL in men or <40 mg/dL in women, and triglycerides >150 mg/dL. Homocysteine levels were considered elevated if they were above 15 nmol/L.

Obesity was defined according to BMI and body fat percentage by bioelectrical impedance analysis. For BMI, the WHO classification was used: grade I obesity (BMI 30–34.9 kg/m^2^), grade II obesity (BMI 35–39.5 kg/m^2^) and grade III obesity (BMI 40–50 kg/m^2^). Obesity according to body fat percentage was >25% body fat in men and >30% in women.

Another anthropometric parameter studied was the final height of the adult patients, according to the percentile charts of the Orbegozo Foundation for the Spanish population over 18 years of age.

Additional nutritional parameters were analyzed such as levels of vitamin B12, folic acid, vitamins A, D and E, iron, albumin and prealbumin, as well as thyroid function by thyroid stimulating hormone and free T4.

Bone mineral health was evaluated by performing bone densitometry (BMD) of the femoral neck and lumbar spine (L1–L4), according to the results indicated in the Nuclear Medicine report and the Z-score data.

The patients were stratified both by degree of metabolic control and type of treatment: exclusive diet and nutrition therapy or treatment with sapropterin. Patients sensitive to sapropterin (BH4) treatment were those who obtained a response of >50% after performing the 24-h sapropterin response test, using the long response test (1 week) in uncertain cases.

Quantitative variables are expressed as median (interquartile range), while qualitative variables are given as percentages (*n* patients). For the comparative cohort study, statistical significance was defined as a value of *p* < 0.05 with higher values considered non-significant. Non-parametric studies were used for the analysis, including the Mann–Whitney U test for independent samples for quantitative variables, and Fisher’s chi-square test for qualitative variables.

## 3. Results

Of the cohort of 90 adult patients with a diagnosis of PKU, 52.2% were women. The rest of the clinical characteristics are shown in Table 1. Regarding the age of the patients, median was 28.5 (23–38) years, and 47.8% (43) were older than 30 years. The age range with the greatest number of patients in the sample was 20–30 years (45.6%), as shown in Figure 1. Patients over 40 years of age accounted for 18.9% (17), and there were no patients over 60 years of age in follow-up.

Of the 62 patients diagnosed by **neonatal screening**, 87.1% (54) were classical forms, 6.5% (4) moderate forms and 6.5% (4) mild forms. Of the 28 patients diagnosed late, 92.9% (26) were classical forms, 3.6% (1) moderate forms and 3.6% (1) mild forms.

Regarding metabolic control of the entire study population, the median Phe was 631 µmol/L (431–863 µmol/L) and tyrosine was 42.2 µmol/L (33.05–54.5 µmol/L). A total of 47.8% (43) of the patients were within the control targets (Figure 2).

In our cohort, 14.6% (12) were **smokers**; 12.4% (10) consumed alcohol and 4.9% (4) consumed other intoxicants. With respect to **physical exercise**, 88.1% (74) described a sedentary lifestyle with little or no physical exercise and 11.9% (10) performed some degree of physical exercise.

The median weight was 71.85 (62–86) kg, with a median height of 165 (158–170) cm. The median height in the men was 170 (167–177) cm with a percentile between P10–P25 (P6–P48) and 159 (154–163) cm in the women with a percentile P25 (P5–P43). Well controlled patients presented a significant higher height than those poorly controlled (168 vs. 163 cm, *p* = 0.022).

The median **BMI** was 26.61 (22.7–31.1) kg/m^2^ with 29.2% (26) of the patients in the obese range, 18% (16) with grade I obesity, 6.7% (6) with grade II obesity, and 4.5% (4) with grade III obesity. Body composition analysis by **bioelectrical impedance** was performed in 35 patients. The median body fat percentage was 24.6% (19.3–31%) and, according to the criteria used, 34% of the patients were obese. An age-associated increase in BMI (*p* = 0.038) was observed with a progressive increase in the prevalence of obesity across the decades: 14.3% (1) between 10–20 years; 14.6% (6) between 20–30 years; 44% (11) between 30–40 years; 50% (6) between 40–50 years; and 80% (4) between 50–60 years. The uncontrolled patients had a significantly higher BMI, with these differences found mainly in the women.

Cardiovascular risk factors are summarized in Table 2.

**Bone mineral density** was analyzed in the lumbar spine (67 patients) and/or in the femoral neck (62 patients). Following the diagnosis according to the BMD report, 48.5% (33) of the patients had normal BMD, 41.17% (28) osteopenia and 10.3% (7) osteoporosis. Concerning the lumbar spine BMDs performed, 50.7% (34) were normal, 38.8% (26) showed osteopenia and 10.4% (7) osteoporosis, with a median Z-score of −1.1 (−1.75–(−0.35)). Of the femoral neck BMDs (62), 64.5% (40) were normal, 32.3% (20) showed osteopenia and 3.2% (2) osteoporosis. The median Z-Score was −0.45 (−1.1–0.55). A significant difference was found with a lower Z-score in lumbar spine densitometry versus femoral neck densitometry (*p* = 0.004). In relation to the plasma levels of **vitamins and micronutrients** studied, vitamin D (25OH) deficiency was noted in 30.6% (26) and vitamin B12 deficiency in 4.5% (4) of our cohort. A trend toward statistical significance (*p* = 0.053) was observed with vitamin B12 deficiency and degree of metabolic control, with a higher percentage of patients with a deficiency in the controlled patients (9.3%) compared to the uncontrolled patients (0%). Analyzing this deficiency with respect to the type of treatment and metabolic control, we observed a 6.9% deficiency in patients controlled with diet compared to a 14.3% deficiency in patients treated with sapropterin. The patients treated with sapropterin had significantly higher homocysteine levels (Table 3).

**Thyroid dysfunction** was present in 6.7% (6) of our adult population with PKU.

### Results According to Metabolic Control

Twenty percent (18/90) of the patients received **sapropterin** (**BH4**) **treatment**, while 80% (72/90) received diet and nutrition therapy. In eight BH4 patients, good metabolic control was not possible with BH4 alone, and they were supplemented with Phe-free amino acid formula. Seventy-seven percent of the patients with BH4 had good metabolic control compared to 38.15% with nutrition therapy; moreover, median Phe levels in patients with BH4 were significantly lower (Figure 2, Table 3).

No significant differences were found in the rest of the parameters evaluated.

## 4. Discussion

The interest of our study lies in the large cohort of adult patients under follow-up in adult units specialized in inborn errors of metabolism. To date, the complications that these patients may experience in the future are not well known. Examining and understanding these complications may therefore help us to prevent future comorbidities.

Our cohort comprised young patients, most of whom were between 20 and 30 years of age. The oldest patient was 56 years old and the youngest 16 years old.

In our series, the proportion of classical PKU was 88.9% compared to 51.5% in other published series. The percentage of patients with moderate PKU (5.6%) or mild PKU (5.6%) was low compared to other series (26% and 15.8%, respectively) [4]. The low representation of mild PKU in our cohort may be explained by losses to follow-up due to older recommendations to discontinue dietary treatment in individuals older than 12 years, or to the genetic characteristics of the population. Nevertheless, further study of the reasons for this low prevalence of mild forms is needed, because if it is a result of loss to follow-up it would be advisable to monitor these patients, especially women of childbearing age, due to the risk of a maternal PKU syndrome and neonatal complications. In our series, 31.1% of the patients were diagnosed late. The indication rate for sapropterin in our cohort was low (20%), probably in relation to the high percentage of patients with classical PKU.

Regarding the **degree of control** in our series, only 40.3% of our patients in diet and nutrition therapy had good metabolic control (according to the recommendations of the European guidelines), while 59.7% were poorly controlled, with Phe levels >600 µmol/L. Within the sapropterin treatment group, the degree of metabolic control was 77.8%, and therefore the total percentage of well-controlled patients in our series was 47.8%, very similar to that of other published series. No significant differences were observed between patients with late diagnosis versus neonatal screening in terms of metabolic control. Adherence to diet and nutrition therapy in adult patients is one of the major challenges to be met in the management of PKU, since high blood Phe concentrations have been associated with cognitive and neuropsychological deterioration in adolescents and adults [6].

Among the **anthropometric** measurements analyzed, the final median height in our cohort was lower than the population values. In both men and women, it was lower than the P50, and this is in agreement with data described in the literature (16). This indicates the importance of the nutritional impact on growth in patients with these types of restrictive diets during childhood.

Obesity, hypertension, diabetes and dyslipidemia, diseases typical of adulthood, are becoming more frequent in our patients. The degree of obesity increases significantly as the age of the sample increases, reaching 80% in patients 50–60 years of age. The prevalence of obesity in our sample was slightly lower than that published in the Andalusian population according to the Di@bet.es Study (29.2% vs. 37%). However, these data are not directly comparable, due to the age difference of the participants in both studies (with a mean age of the population, from which the Andalusian prevalence was extrapolated, of 48.5 years). Other studies (European Health Survey 2020 and National Health Survey 2017) establish a much lower prevalence of obesity in the age range of our cohort, reaching up to 16.2% in men and 12.9% in women in the age range of 35 to 44 years [5,7].

The studies by Azabdaftari et al. and de Almeida et al. reported associations between BMI and Phe control. Thus, poorly controlled patients had a significantly higher BMI than controlled patients, which can have a significant influence on low-density lipoprotein (LDL), HDL, and total cholesterol levels. In the study by de Almeida et al., however, the correlation between weight and Phe levels was not controlled for age, a factor associated with weight gain, and a correlation was also found between Phe levels and age [8]. In our study, although there were differences in BMI between controlled and uncontrolled patients, these differences were mainly due to those found in the women (with an absolute difference in BMI of more than 5 points), with a significantly higher number of women with a BMI > 35 in the uncontrolled group. It would be necessary to analyze current and past dietary patterns as well as lifestyle and cognitive situation to develop possible reasons for this difference. Nonetheless, we believe that a specific assessment is needed in PKU patients, not only of weight but also of body composition. Current **body composition** assessment techniques allow an accurate diagnosis of fat mass percentage and an assessment of visceral fat. A sample of our series underwent bioelectrical impedance analysis, which showed increased sensitivity in the diagnosis of obesity through this technique (obesity according to BMI 29.2% vs. 34.3% obesity according to body fat percentage, *p* = 0.001). It is possible that the specific diet followed by these patients, without natural proteins and rich in carbohydrates, influences not only weight but also body composition. Preliminary studies have suggested an independence between body composition and dietary treatment [9]. Later studies have suggested the possibility of a higher fat percentage in these patients at an older age, and with a longer period of time on the diet [10].

In terms of **lifestyle**, the high percentage of sedentary patients in the sample is significant (88.1%), and is associated with both age (*p* = 0.027) and BMI (*p* = 0.011), as well as with a high consumption of toxic substances (13.4%).

Regarding **vascular comorbidities**, in gross comparison with the Spanish population, we highlight the low frequency of hypertension with a reference prevalence according to the Di@bet.es study of 42.6% of the population [5]. It is true that this prevalence is based on a population with a mean age of 50.5 years. However, stratifying by age and using groups similar to the standardized prevalence determined in the study (18–30 years, 31–45 years, and 45–60 years), we observed a lower prevalence of hypertension in all three groups: 18–30 years (2% vs. 9.3%), 31–45 years (13.3% vs. 17.2%), and 46–60 years (22.2% vs. 44.4%). Several studies have associated PKU with a higher BMI, a higher prevalence of hypertension, and a worse lipid profile (triglycerides, LDL) in the most poorly controlled patients [11]. In our study, no significant differences in the cardiovascular risk variables recorded between controlled and uncontrolled patients were observed.

Limited evidence is available on PKU and increased cardiovascular risk, both as a consequence of the disease itself, and of the intrinsic characteristics of a diet restricted in natural proteins. Some studies have shown that PKU patients have a higher vascular risk compared to healthy controls [10], with a higher BMI, a higher prevalence of hypertension, higher levels of total and non-HDL cholesterol, lower levels of HDL cholesterol, greater carotid stiffness, a worse inflammatory profile, and higher plasma levels of C-reactive protein and amyloid A protein as well as markers of oxidative stress [11]. Nevertheless, other studies that have used metabolomics to study the lipid profile have found lower plasma levels of total cholesterol and LDL cholesterol compared to healthy controls [12]. This variability in results may be associated with adherence to diet and/or supplementation, as other studies have shown [6].

**Vitamin B12 and homocysteine** levels are significantly different between patients receiving sapropterin treatment and those following a restricted diet. The explanation for this difference is that Phe-free amino acid supplements are already fortified with vitamin B12; patients on sapropterin treatment, although they usually follow an unrestricted diet, do not have as balanced a diet, especially in animal proteins, as the healthy population. Despite unrestricted diets, dietary and nutritional monitoring of these patients is very important to detect deficiencies.

**Bone mineral health** is an important area, requiring explicit follow-up in PKU patients. In classical studies [7], abnormalities in bone development have been observed, although the results are inconclusive given the rarity of the disease and the number of pediatric patients included in the studies (peak bone mass occurs between the ages of 25 and 35 years). PKU is not currently classified as a cause of secondary osteoporosis, unlike inborn errors of metabolism such as homocystinuria. However, it has been reported that PKU patients have lower bone mineral density levels than healthy controls, and there appears to be a higher rate of fractures in these patients [8]. Observational studies have raised the possibility that decreased natural protein intake as well as BMI may be factors influencing the development of this comorbidity [9]. The vitamin D deficiency observed in our cohort is similar to that published at the population level (30.6% vs. 33.9%) [12]. The addition of vitamins and micronutrients to hydrolyzed formulas of Phe-free amino acids may mediate the risk of osteopenia/osteoporosis and vitamin D deficiency. For this reason, bone mineral density abnormalities may not be as common as in the past.

Our bone mineral density data are consistent with published data. In our series, we have no record of fragility fractures, but a high number of patients diagnosed with osteoporosis (10.3%) were observed, taking into account the median age of the sample. In the studies published to date, the bone mineral density of PKU patients was significantly lower than that of healthy controls, although generally within a normal range with respect to the reference range. The variability of the bone mineral density comparison and the use of the Z-score or T-score in the different published studies makes it difficult to make an adequate comparison [2]. Different studies establish the prevalence of densitometric osteoporosis in women at 20–44 years in 0.34% (lumbar spine) and 0.17% (femoral neck) and for men at the same age in 1.39% and 0.17%, respectively [13]. In our opinion, although additional studies are needed to analyze bone health through different techniques and methods (BMD, logarithms of long-term fracture risk or trabecular bone score), we believe that PKU should be introduced into diagnostic algorithms as a cause of secondary osteoporosis. Similarly, due to the characteristics of the disease and its etiopathogenesis, a consensus is also required to establish treatment algorithms and the age at which the initiation of treatment optimizes the benefits while minimizing the potential risks.

Apart from the better metabolic control achieved in patients treated with sapropterin, we found no statistically significant differences in the anthropometric variables and metabolic comorbidities studied in our cohort.

## 5. Conclusions

Our study contributes a large cohort of adult patients with PKU under close follow-up. The initiation of neonatal screening together with improved and earlier treatments ensure that these patients have better cognitive and social function than observed in classical cohorts. These patients develop comorbidities specific to these older ages that were not previously known, and with certain singularities associated with their PKU, which makes it necessary to publish a large series of adult patients to study these comorbidities.

In our series, we observed no significant differences with most of the vascular risk factors. We underscore the difference in the prevalence of obesity between the PKU population under follow-up and the age-adjusted Spanish prevalence in our cohort. We believe that body composition should be evaluated due to the low sensitivity of BMI for the detection of obesity in these patients.

This study highlights the importance of closely monitoring vitamins, micronutrients, and bone mineral health in the context of the disease and its treatment.

It is vitally important to underscore the need for lifelong follow-up in PKU patients, even in those with mild forms, as well as to emphasize healthy lifestyle recommendations based on a comprehensive assessment.

## Figures and Tables

**Figure 1 nutrients-14-01311-f001:**
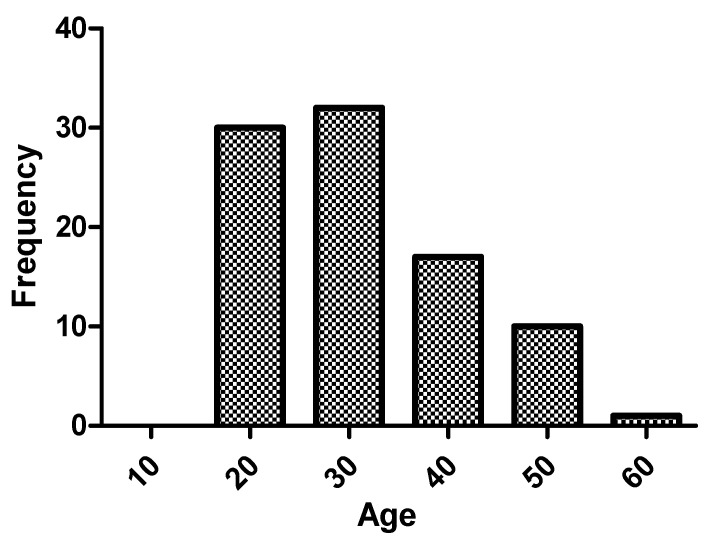
Histogram of frequency by age.

**Figure 2 nutrients-14-01311-f002:**
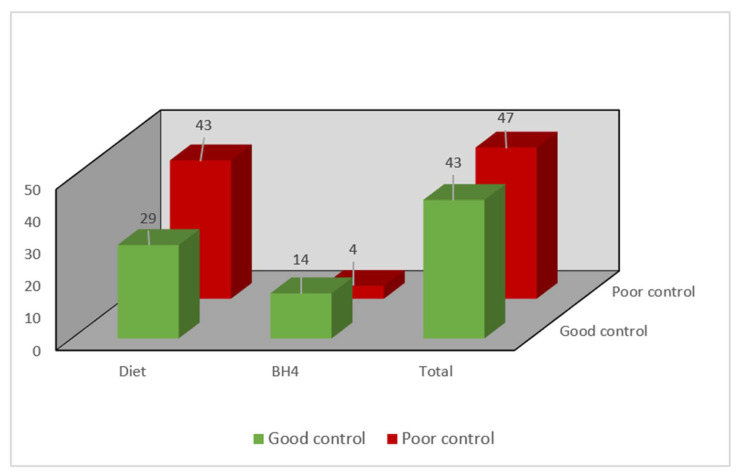
Number of patients (*n*) according to metabolic control. BH4: tetrahydro-biopterin.

**Table 1 nutrients-14-01311-t001:** Characteristics of the cohort of PKU patients.

Number of Patients	*n* = 90
Sex (*n*, %)	
Women	47 (52.2%)
Men	43 (47.8%)
Mean age (years)	29 (23–38)
Minimum age	16
Maximum age	56
PKU type (*n*, %)	
Classical	80 (88.9%)
Moderate	5 (5.6%)
Mild	5 (5.6%)
Diagnosis	
Neonatal Screening	62 (68.9%)
Late	28 (31.1%)

PKU: Phenylketonuria.

**Table 2 nutrients-14-01311-t002:** Cardiovascular risk factors.

	Patients (%, *n*)	Plasma Levels
Hypertension	7.9% (7)	
Type 2 diabetes mellitus	2.2% (2)	
Dyslipidemia		
-Hypercholesterolemia (total cholesterol)	15.6% (14)	159 (138–177) mg/dL
-Lowered HDL cholesterol	15.8% (9)	47 (41–53) mg/dL
-Hypertriglyceridemia	17.8% (16)	92 (68–122) mg/dL
Hyperhomocysteinemia	18.2% (12)	9.49 (7.3–13.8) nmol/L
Obesity		
Body mass index	29.2% (26)	
Bioelectrical impedance analysis	34% (12)	

Qualitative variables represented as % (*n*) of patients affected. Quantitative variables represented as median (p25–p75). Patients were considered to have hypercholesterolemia when total cholesterol was >200 mg/dL, HDL < 35 mg/dL in men or <40 mg/dL in women, and triglycerides >150 mg/dL. Elevated homocysteine levels were defined as >15 nmol/L.

**Table 3 nutrients-14-01311-t003:** Clinical differences between controlled and uncontrolled patients (according to Phe levels).

	Controlled (43)	Uncontrolled (47)	*p*
Sapropterin	32.6% (14)	8.5% (4)	0.07
Treatment			
Diet	67.4% (29)	91.5% (43)	
BMI			
Total	24.36 (21.75–29.41) kg/m^2^	27.45 (24.14–32.81) kg/m^2^	0.023
Women	22.58 (21.18–28.33) kg/m^2^	28.11 (26.04–33.71) kg/m^2^	0.007
Men	26.3 (23.77–30.02) kg/m^2^	26.5 (23.51–30.25) kg/m^2^	0.923
Vitamin B12	424 (308–801) pg/mL	530 (419–751) pg/mL	0.347
	9.3% (4) *	0% (0)	0.053

Qualitative variables represented as % (*n*) of patients in the category. Quantitative variables represented as median (p25–p75). * % (*n*) of patients with diagnosed vitamin B12 deficiency.

## Data Availability

Data available on request.

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
