# Peer review of "Cardiometabolic and Nutritional Morbidities of a Large, Adult, PKU Cohort from Andalusia"

_nutrients, 2022, doi:10.3390/nu14061311_

Round 1

Reviewer 1 Report

This article described the nutritional status and cardiovascular risk factors in a cohort of patients with PKU from Andalusia from two metabolic centers. They nicely categorized by treatment status and compared to reference populations. They noted a relative drawback was the younger age of the population.

I thought the paper was excellent, well written and translated, and provided appropriate level of detail in describing the population.

Suggestions and questions raised:

I wasn’t certain from the abstract if the numbers in brackets after median age was the age range or standard deviation, but this was clarified later in the paper. Perhaps add “SD” in the brackets for clarification.

In considering the age of the cohort and the percent diagnosed by NBS, I wanted to know when newborn screening began and was mandatory in your region.

I was surprised by the degree of short stature in the population under study. Was there a difference in early and late treated for stature and we’ll controlled vs poorly controlled?

Did those on BH4 overlap with those on formula? This was unclear. Is the BH4 cohort referring to BH4 with or without formula/diet?

In figure 2 I would suggest replacing “bad control” with “poor control”. In table 3 it may be helpful to compare to BMI in non-PKU population as well.

On page 7, I wasn’t completely clear but assuming the Diabetes study figures you used were for normal controls without diabetes? Perhaps state if that was a non-diabetic general population unless I missed it.

Why do you think the poorly controlled (those more likely on a typical unrestricted diet) had higher BMI? From dietary habits from being on diet in the past (carb heavy, no meat) or were the late treated cognitively different and with less knowledge of healthy eating or less access to healthy foods, or other reasons? This may be interesting to elaborate on.

When you refer to toxic substances, I believe this refers to alcohol and smoking. Was there a threshold for alcohol quantity or frequency because it is surprising only 12% drink alcohol if you are including social drinking.

In the final paragraph of the discussion section, consider as a comparison to give the rate of osteoporosis in the general population of similar age.

In the discussion of osteoporosis you may consider that the addition of vitamins/micronutrients to metabolic formulas may mediate the risk of osteopenia/osteoporosis. For that reason the problem may be less now than it was in the past when formulas were not as well fortified.

In your conclusions I wanted to ask about the statement about longer survival. PKU itself is not life limiting to my understanding, but patient may have died younger due to institutionalization and related social constructs. Neurodegeneration is not a given though there have been case reports.

We just have less experience with older individuals because many are undiagnosed, and the natural history in the treated state is a story currently unfolding.

In the second to last paragraph I question the use of the word “deterioration”associated with the disease and treatment.

Overall excellent work and thank you for this great contribution to the field!

Author Response

This article described the nutritional status and cardiovascular risk factors in a cohort of patients with PKU from Andalusia from two metabolic centers. They nicely categorized by treatment status and compared to reference populations. They noted a relative drawback was the younger age of the population.

I thought the paper was excellent, well written and translated, and provided appropriate level of detail in describing the population.

Response: Thank you for your words, It´s been a really big work and a combined effort between two large centers

Suggestions and questions raised:

I wasn’t certain from the abstract if the numbers in brackets after median age was the age range or standard deviation, but this was clarified later in the paper. Perhaps add “SD” in the brackets for clarification.

Response: As described later in the "materials and methods" section, quantitative variables are expressed as the median [intercuartilic range]. Nevertheless, we have modified abstract in order to clarify this question.

In considering the age of the cohort and the percent diagnosed by NBS, I wanted to know when newborn screening began and was mandatory in your region.

R: In Andalusia, newborn screening for hypothyroidism and phenylketonuria began and was mandatory from 1978.

I was surprised by the degree of short stature in the population under study. Was there a difference in early and late treated for stature and we’ll controlled vs poorly controlled?

R: There is a significant difference in stature between well and poorly controled. This has been added to the manuscript: "Well controlled patient presented a significant higher height than those poorly controlled (168 Vs 163 cm, p=0.018)."

This difference hasn´t seen between early and late diagnosis or BH4 treatment.

Did those on BH4 overlap with those on formula? This was unclear. Is the BH4 cohort referring to BH4 with or without formula/diet?

R: Some patients need formula plus BH4, althought it is not frequent in our cohort. 8/18 patient recieve specific formula added to BH4 treatment.

In figure 2 I would suggest replacing “bad control” with “poor control”. In table 3 it may be helpful to compare to BMI in non-PKU population as well.

R: The term in figure 2 has been changed. Unfortunately, we don´t have specific BMI data from controlled non PKU patients.

On page 7, I wasn’t completely clear but assuming the Diabetes study figures you used were for normal controls without diabetes? Perhaps state if that was a non-diabetic general population unless I missed it.

R: Di@bet.es study was a population-based, cross-sectional, cluster sampling study with target population being the entire Spanish population. 5072 participants were randomly selected in order to get hypertension, diabetes mellitus and obesity prevalence

Why do you think the poorly controlled (those more likely on a typical unrestricted diet) had higher BMI? From dietary habits from being on diet in the past (carb heavy, no meat) or were the late treated cognitively different and with less knowledge of healthy eating or less access to healthy foods, or other reasons? This may be interesting to elaborate on.

R: We think all these reasons may be plausible. The explanation for this reality is beyond the scope of the present study. Indeed, we are developing more detailed studies on diet and cognnition to explore this difference among other situations.

When you refer to toxic substances, I believe this refers to alcohol and smoking. Was there a threshold for alcohol quantity or frequency because it is surprising only 12% drink alcohol if you are including social drinking.

R: Our dietist asks about every alcohol consumption with no threshold. And every alcohol consumption was considered.

In the final paragraph of the discussion section, consider as a comparison to give the rate of osteoporosis in the general population of similar age.

R: We don´t have age-specific prevalence for osteoporosis in our population. Secondary osteoporosis in young population have been estimed in 0.5% (specifically, different studies establish the prevalence of densitometric osteoporosis in women at 20-44 years in 0.34% (lumbar spine) and 0.17% (femoral neck) and for men at the same age in 1.39% and 0.17%, respectively [13]. 

In the discussion of osteoporosis you may consider that the addition of vitamins/micronutrients to metabolic formulas may mediate the risk of osteopenia/osteoporosis. For that reason the problem may be less now than it was in the past when formulas were not as well fortified.

R: Indeed. We have added this to discussion.

In your conclusions I wanted to ask about the statement about longer survival. PKU itself is not life limiting to my understanding, but patient may have died younger due to institutionalization and related social constructs. Neurodegeneration is not a given though there have been case reports.

We just have less experience with older individuals because many are undiagnosed, and the natural history in the treated state is a story currently unfolding.

R: Indeed, we have changed this sentence.

In the second to last paragraph I question the use of the word “deterioration”associated with the disease and treatment.

R: It has been changed

Overall excellent work and thank you for this great contribution to the field!

Reviewer 2 Report

In this study, the authors described the demographics of PKU cohort.

This has an important clinical meaning for PKU research field.

Author Response

In this study, the authors described the demographics of PKU cohort.

This has an important clinical meaning for PKU research field.

Thank you for your kind words. We make our best efforts